# GAS5 protects against osteoporosis by targeting UPF1/SMAD7 axis in osteoblast differentiation

Ming Li[1,2†], Zhongyu Xie[1†], Jinteng Li[1,2†], Jiajie Lin[1], Guan Zheng[1,2], Wenjie Liu[1,2], Su'an Tang[3], Shuizhong Cen[2,3], Guiwen Ye[2], Zhaofeng Li[2], Wenhui Yu[1], Peng Wang[1]*, Yanfeng Wu[4,5]*, Huiyong Shen[1,2]*

[1]Department of Orthopedics, The Eighth Affiliated Hospital, Sun Yat-sen University, Shenzhen, China; [2]Department of Orthopedics, Sun Yat-sen Memorial Hospital, Sun Yat-sen University, Guangzhou, China; [3]Department of Orthopedics, Zhujiang Hospital, Southern Medical University, Guangzhou, China; [4]Center for Biotherapy, The Eighth Affiliated Hospital, Sun Yat-sen University, Shenzhen, China; [5]Center for Biotherapy, Sun Yat-sen Memorial Hospital, Sun Yat-sen University, Guangzhou, China

**Abstract** Osteoporosis is a common systemic skeletal disorder resulting in bone fragility and increased fracture risk. It is still necessary to explore its detailed mechanisms and identify novel targets for the treatment of osteoporosis. Previously, we found that a lncRNA named *GAS5* in human could negatively regulate the lipoblast/adipocyte differentiation. However, it is still unclear whether *GAS5* affects osteoblast differentiation and whether *GAS5* is associated with osteoporosis. Our current research found that *GAS5* was decreased in the bones and BMSCs, a major origin of osteoblast, of osteoporosis patients. Mechanistically, *GAS5* promotes the osteoblast differentiation by interacting with UPF1 to degrade *SMAD7* mRNA. Moreover, a decreased bone mass and impaired bone repair ability were observed in *Gas5* heterozygous mice, manifesting in osteoporosis. The systemic supplement of *Gas5*-overexpressing adenoviruses significantly ameliorated bone loss in an osteoporosis mouse model. In conclusion, *GAS5* promotes osteoblast differentiation by targeting the UPF1/SMAD7 axis and protects against osteoporosis.

*For correspondence:
wangpengsmh@foxmail.com (PW);
wuyf@mail.sysu.edu.cn (YW);
shenhuiy@mail.sysu.edu.cn (HS)

†These authors contributed equally to this work

Competing interests: The authors declare that no competing interests exist.

## Introduction

Osteoporosis is a systemic skeletal disease that manifests as low bone mass and microarchitectural deterioration of bone tissue (*Yang et al., 2020*). Individuals with osteoporosis are at a higher risk of fragility fractures (*Eastell et al., 2016*). Over 200 million people worldwide suffer from this disease, and this number is gradually increasing (*Lizneva et al., 2018*). Fractures caused by osteoporosis lead to functional disruption and pain along with an enormous therapeutic cost. In the United States and European Union, osteoporotic fractures have been estimated to cost $20 billion to $30 billion annually (*Lorentzon, 2019*). Although many studies on osteoporosis have been conducted in recent years, more effective diagnostic and curative targets need to be explored for osteoporosis.

The imbalance between bone and fat mass is a typical feature of the osteoporosis pathogenesis (*de Paula and Rosen, 2020*; *de Paula and Rosen, 2017*; *Infante and Rodríguez, 2018*). Recent studies have demonstrated that the abnormal differentiation capacities of bone marrow stromal cells (BMSCs) play an important role in this critical pathogenesis of osteoporosis (*Li et al., 2019*; *Ma et al., 2018*). BMSCs are stem cells that possess self-renewal and multiple differentiation abilities (*Grayson et al., 2015*). As the major origins of osteoblasts and adipocytes, BMSCs maintain their balance between osteoblast and adipocyte, which are regulated by

various molecular checkpoints (*Anthony and Link, 2014*; *Williams and Hare, 2011*). The abnormal expression of these molecules disrupts the osteoblast and adipocyte differentiation balance and leads to disordered bone metabolism, ultimately resulting in disease development (*Xu et al., 2018*; *Zheng et al., 2020*). Therefore, identifying these critical targets of osteoblast and adipocyte differentiation will be significantly helpful for the early diagnosis and treatment of diseases.

Long noncoding RNAs (lncRNAs) are a type of RNA longer than 200 nt and do not have protein-coding potential (*Marchese et al., 2017*). The various and vital functions of lncRNAs in cell biology have been proven in numerous studies (*Tang et al., 2017*; *Yao et al., 2019*). Previous studies have revealed that lncRNAs are involved in BMSCs osteoblast and adipocyte differentiation (*Xie et al., 2019*). *GAS5* is a well-researched lncRNA that widely contributes to various cell biology functions such as immunity, tumor generation and metastasis (*Renganathan et al., 2014*; *Sun et al., 2017*; *Wang et al., 2016*). In our previous study, we demonstrated that *GAS5* negatively affected the adipoytes differentiation through the *miR-18a*/CTGF pathway (*Li et al., 2018*). Other researchers also recognized the crucial role of *GAS5* in BMSCs differentiation. For example, the researches of Wang, X etc. indicated that *Gas5* could promotes osteogenic differentiation of bone marrow mesenchymal stem cells by FOXO1 in mouse (*Wang et al., 2019*).However, the effect of *GAS5* on human osteoblast differentiation and its role in osteoporosis have not been clearly clarified.

In our current research, we found that the level of *GAS5* was decreased in both the bones and BMSCs of osteoporosis patients. Molecular and in vivo studies revealed that *GAS5* promoted the osteoblast differentiation through the UPF1/SMAD7 axis. Furthermore, *Gas5* heterozygous mice showed decreased bone mass and an impaired bone repair capacity as an osteoporotic phenotype. Bone loss was found to be alleviated in an osteoporosis mouse model after systemic application with *Gas5*-overexpressing adenoviruses. These findings consistently support the hypothesis that the lncRNA GAS5 acts as a protective target in osteoporosis by regulating osteoblast differentiation.

## Results

### GAS5 decreased in osteoporosis and positively correlated with the osteoblast differentiation

Eight osteoporosis patients who suffered from femoral neck fractures and eight patients with hip dysplasia and eight normal controls with car accident requiring surgery were recruited (*Supplementary file 3*). BGLAP and COL1A1 immunohistochemical staining confirmed the decrease in the bone formation marker in osteoporosis patients (*Figure 1A*). The mRNA level of *GAS5* in both bone tissues and BMSCs was decreased in patients with osteoporosis compared to in those with hip dysplasia and normal control (*Figure 1B–C*), which indicated that *GAS5* expression was closely related to bone metabolism in osteoporosis. Given the crucial role of BMSCs in osteoporosis, we explored the variation tendency of *GAS5* expression during osteoblast differentiation of BMSCs. The results showed that *GAS5* was upregulated along with the induction of osteogenesis, as shown by ARS assays, ALP staining, ALP tests and qPCR (*Figure 1D–E*, *Figure 1—figure supplement 1*). In addition, there was a positive relationship between the expression of *GAS5* and the quantification of ARS staining (*Figure 1F*) or ALP activity (*Figure 1—figure supplement 2A*) or *RUNX2/COL1A11/ BGLAP* mRNA expression (*Figure 1—figure supplement 2B–D*) in osteoblast differentiation. The distribution of lncRNAs is crucial for their function in cell behavior. As shown in *Figure 1G*, the distribution of *GAS5* was more abundant in the cytoplasm. This tendency remained unchanged after induction (*Figure 1H–I*).

### Decreasing GAS5 inhibited the osteoblast differentiation in vitro and in vivo

To explore the function of *GAS5* in osteoblast differentiation, two siRNAs with better knockdown efficiency were chosen (*Figure 2A*) to construct the *GAS5*-knockdown lentiviruses (shRNA-1, shRNA-2). Decreased osteoblast differentiation was shown in the ARS assay (*Figure 2B*) and ALP assay (*Figure 2C*) after decreasing *GAS5* expression. The expression of osteogenesis markers, including *RUNX2*, *COL1A1*, *BGLAP*, *ALPL*, and *SP7*, showed consistent results at both the mRNA (*Figure 2D*) and protein levels (*Figure 2—figure supplement 1A–B*). Then, we constructed a in vivo bone formation experiment to determine the role of *GAS5* in osteoblast differentiation. The new bone

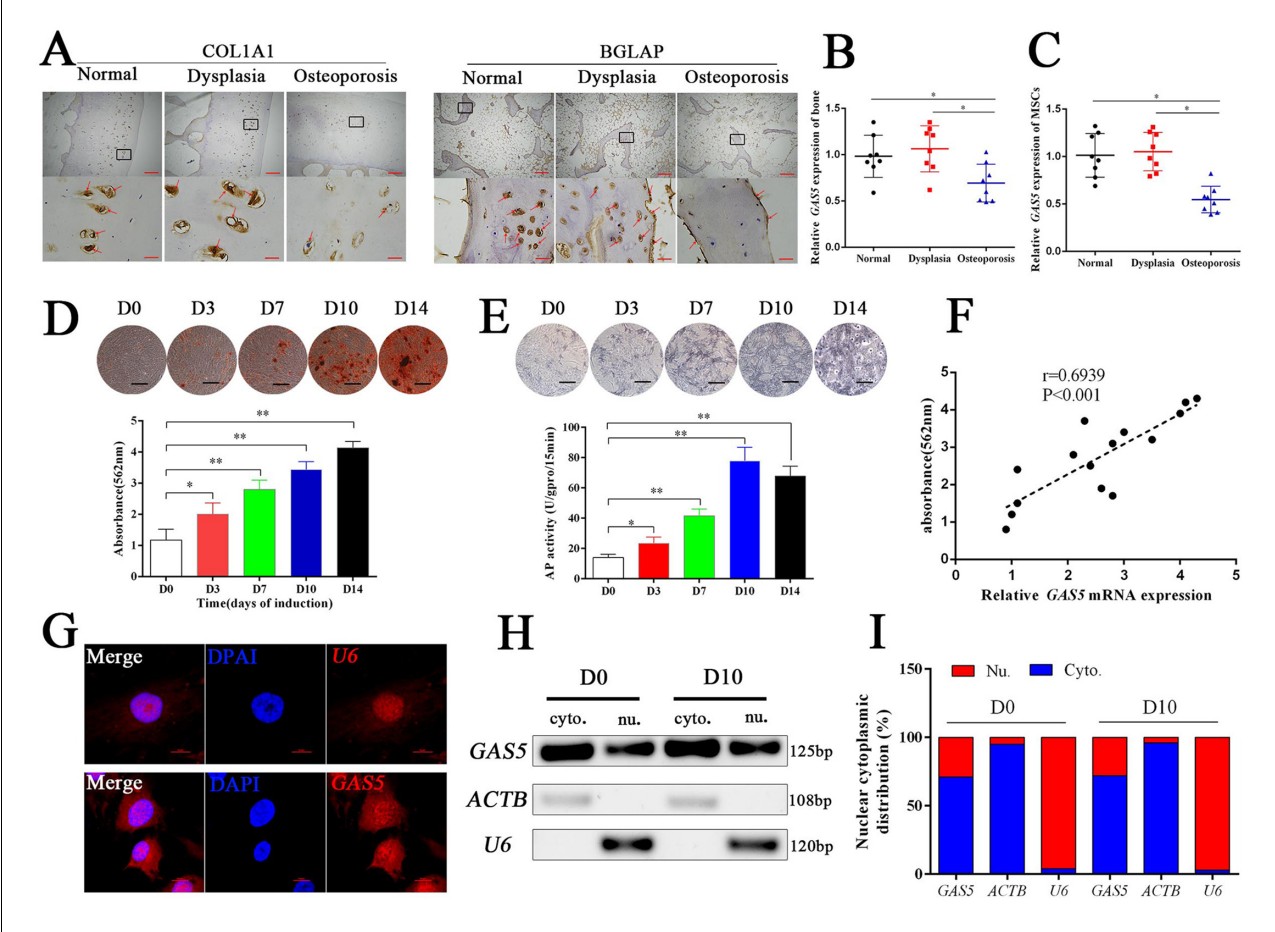

**Figure 1.** *GAS5* decreased in osteoporosis and positively correlated with the osteoblast differentiation. (**A**) COL1A1 and BGLAP immunohistochemical staining of the femur head from hip dysplasia and osteoporosis patients. The red arrow shows typical staining. Scale bar, (upper, 500 μm; lower, 50 μm). (**B**) qRT-PCR analysis of *GAS5* isolated from cancellous bone between eight patients with postmenopausal osteoporosis and eight patients with hip dysplasia (n = 8). (**C**) *GAS5* mRNA expression level in BMSCs isolated from hip dysplasia and osteoporosis patients (n = 8). (**D**) ARS staining during the osteogenic differentiation of MSCs (top). Alizarin red staining quantification during the osteogenic differentiation of MSCs. (bottom) (n = 15). (**E**) ALP staining and ALP assays in the osteoblast differentiation. (**F**) The relation of *GAS5* and Alizarin red staining quantification in the osteoblast differentiation (n = 15). (**G**) *GAS5* RNA FISH in BMSCs. *U6* as the positive control. Scale bar, 50 μm. (**H, I**) Nuclear and cytoplasmic fractionation assay following agarose gel electrophoresis and analysis of *GAS5* in the osteoblast differentiation. ACTB (actin β) and U6 were used as the positive controls. The online version of this article includes the following figure supplement(s) for figure 1:

**Figure supplement 1.** Gene marker expression of BMSCs osteoblast differentiation.

**Figure supplement 2.** The relation of *GAS5* and the other makers of MSCs in the osteoblast differentiation.

formation shown by H and E, Masson staining and BGLAP immunohistochemical staining in the shGAS5 group was significantly lower than that in the control group (*Figure 2E*, *Figure 2—figure supplement 2*).

## Increasing GAS5 promotes osteogenesis in vitro and in vivo

The expression level of *GAS5* in BMSCs was greatly increased in the OE-GAS5 (overexpression of GAS5) group (*Figure 3A*). After transfection with OE-GAS5, the osteoblast differentiation was significantly promoted, as determined by both ARS assays (*Figure 3B*) and ALP assays (*Figure 3C*). Similar results were confirmed by osteoblast marker expression at both the gene and protein levels (*Figure 3D*). Moreover, increased bone formation in the OE-GAS5 group was observed in the bone formation model by H and E staining, Masson staining and BGLAP immunohistochemical staining (*Figure 3E*, *Figure 3—figure supplement 1*). These results indicated that *GAS5* positively promoted the osteoblast differentiation in vitro and in vivo.

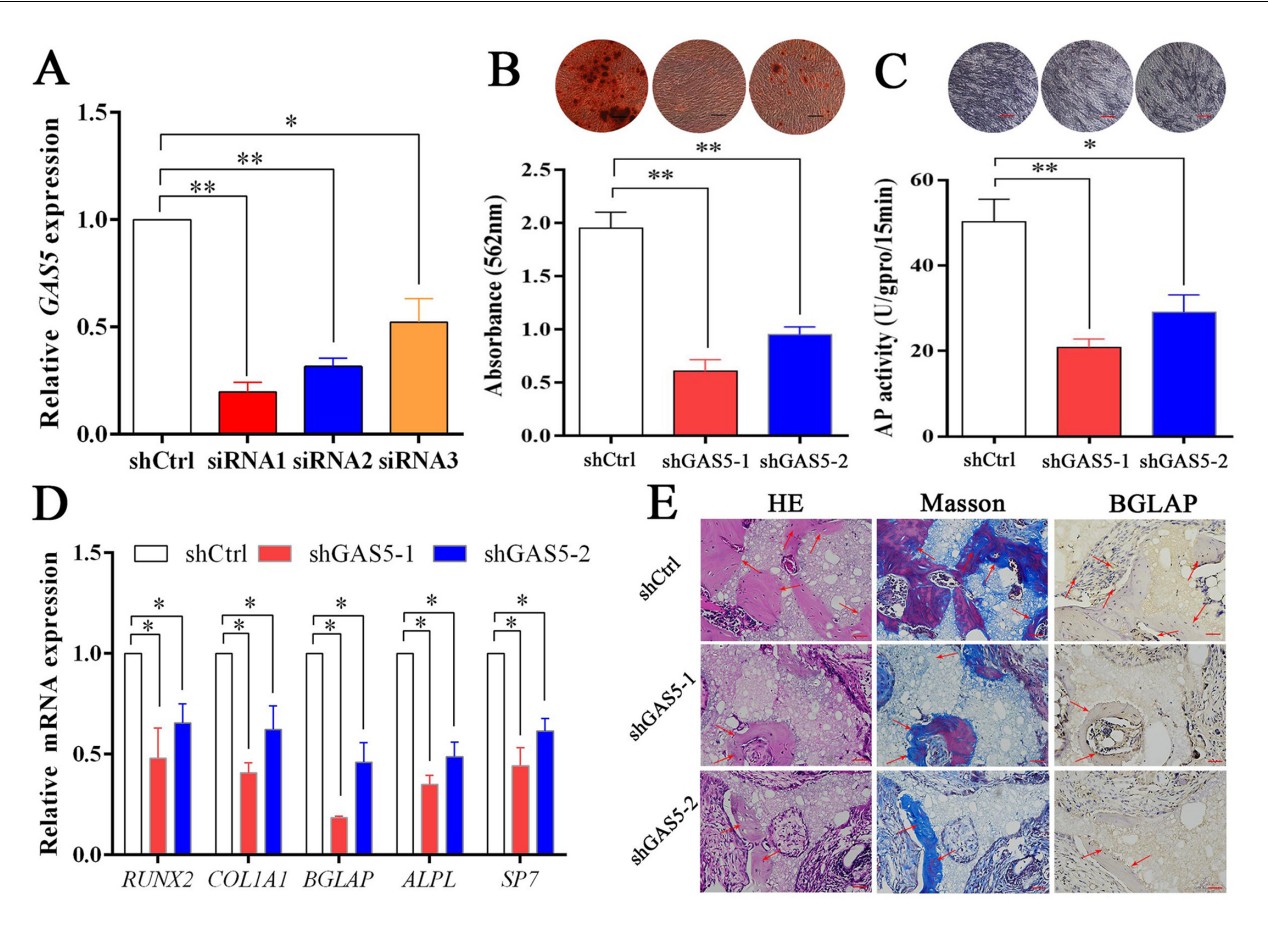

**Figure 2.** Decreasing *GAS5* inhibited the osteoblast differentiation in vitro and in vivo. (**A**) *GAS5* siRNA knockdown efficiency tested by qRT-PCR. (**B**) ARS staining and quantification in the *GAS5* knockdown or control group. Scale bar, 500 μm. (**C**) ALP staining and ALP tests in the *GAS5* knockdown or control group. Scale bar, 500 μm. (**D**) Relative *RUNX2*, *ALPL*, *BGLAP*, *COL1A1*, and *SP7* expression in the *GAS5* knockdown or control group. (**E**) H and E staining, Masson staining and *Bglap* immunohistochemical staining of HA/TCP in the *GAS5* knockdown or control group. The red arrow shows typical bone formation. Scale bar, 100 μm; n = 5. Each cellular experimental group was repeated at least three times.

The online version of this article includes the following figure supplement(s) for figure 2:

**Figure supplement 1.** Western blot analysis and quantification of osteogenesis markers.

**Figure supplement 2.** The quantification of BGLAP immunohistochemical staining.

## GAS5 functions by interacting with the UPF1 protein

To further investigate the detailed mechanism of *GAS5* in osteoblast differentiation, we first detected the adjacent genes of *GAS5* as in previous studies. The results showed that no significant changes in the adjacent genes were observed after inhibiting *GAS5* expression, indicating that it did not function through cis regulation (*Figure 4—figure supplement 1*). The lncRNA-protein interaction is also an important aspect of its function (*Marchese et al., 2017*). Then, we performed an RNA pull-down assay to identify the interacting protein of *GAS5*. The mass spectrometry results indicated that UPF1, a DNA/RNA helicase at the crossroads of many critical cellular pathways for RNA and DNA maintenance (*Fiorini et al., 2018*), was identified in the *GAS5* pull-down assay (*Figure 4A*). Western blot analysis of the *GAS5* pull-down protein confirmed that *GAS5* could interact with UPF1 specifically (*Figure 4B*), and a RIP assay using the anti-UPF1 antibody further confirmed this conclusion (*Figure 4C–D*). In addition, the locations of *GAS5* and UPF1 in BMSCs partly overlapped, further indicating the interaction of *GAS5* and UPF1 (*Figure 4E*).

To further illuminate the specific interaction sites of the *GAS5*/UPF1 complex, we constructed *GAS5* mutant RNA according to the structure and sequence of *GAS5* (*Figure 4F*, *Figure 4—figure supplement 2*). RNA pull-down followed by western blot analyses showed that *GAS5 (210–420)* was

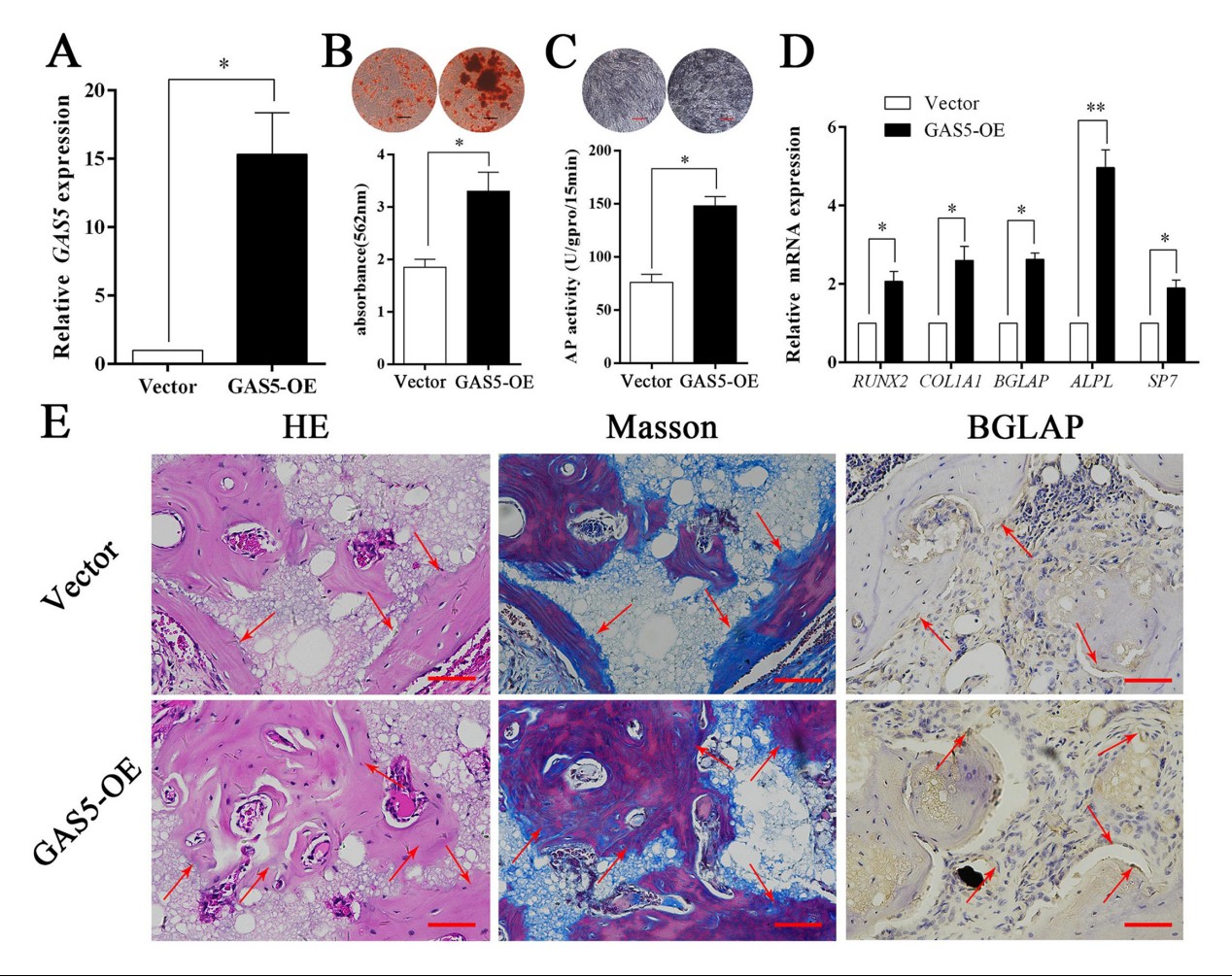

**Figure 3.** Increasing *GAS5* in BMSCs promotes osteoblast differentiation in vitro and in vivo. (A) Relative *GAS5* expression in the overexpression and control groups. (B) ARS staining and quantification in the *GAS5* overexpression or control group. Scale bar, 50 μm. (C) ALP staining and ALP tests in the *GAS5* overexpression or control group. (D) Relative *RUNX2, ALP, BGLAP, COL1A1,* and *SP7* expression in the *GAS5* overexpression or control group. (E) H and E, Masson staining and BGLAP immunohistochemical staining of HA/TCP in the *GAS5* overexpression or control group. The red arrow shows typical bone formation. Scale bar, 100 μm; n = 5. Each cellular experimental group was repeated at least three times.

The online version of this article includes the following source data and figure supplement(s) for figure 3:

**Source data 1.** Original images of alizarin red staining and ALP staining in experiments of 'GAS5 interacts with UPF1 to accelerate SMAD7 decay'.
**Figure supplement 1.** The quantification of BGLAP immunohistochemical staining.

indispensable in the interaction with UPF1 and *GAS5* was unable to detect UPF1 when this sequence was lost (*Figure 4G*). UPF1 protein has been found to include three different functional regions, including CH, HD, and SQ regions (*Dehecq et al., 2018*; *Fiorini et al., 2018*). Based on the structure of UPF1, we constructed the Flag-tagged UPF1 truncation plasmid (*Figure 4H*), and RIP assays were performed to explore the *GAS5* interaction sites in UPF1. The results showed that UPF1-HD specifically combined with *GAS5* (*Figure 4I–J*), which indicated that UPF-HD is indispensable in the interaction of *GAS5* and UPF1.

## GAS5 interacts with UPF1 to accelerate SMAD7 decay

To further explore how *GAS5* interacted with UPF1 to regulate the osteoblast differentiation, the activation levels of three critical signaling pathways, including the TGF-β/SMAD1/5/8, catenin and ERK pathways, were analyzed. As shown in *Figure 5A* and *Figure 5—figure supplement 1A–C*, the SMAD1/5/8 signaling pathway was significantly decreased in the shGAS5 group but increased in the

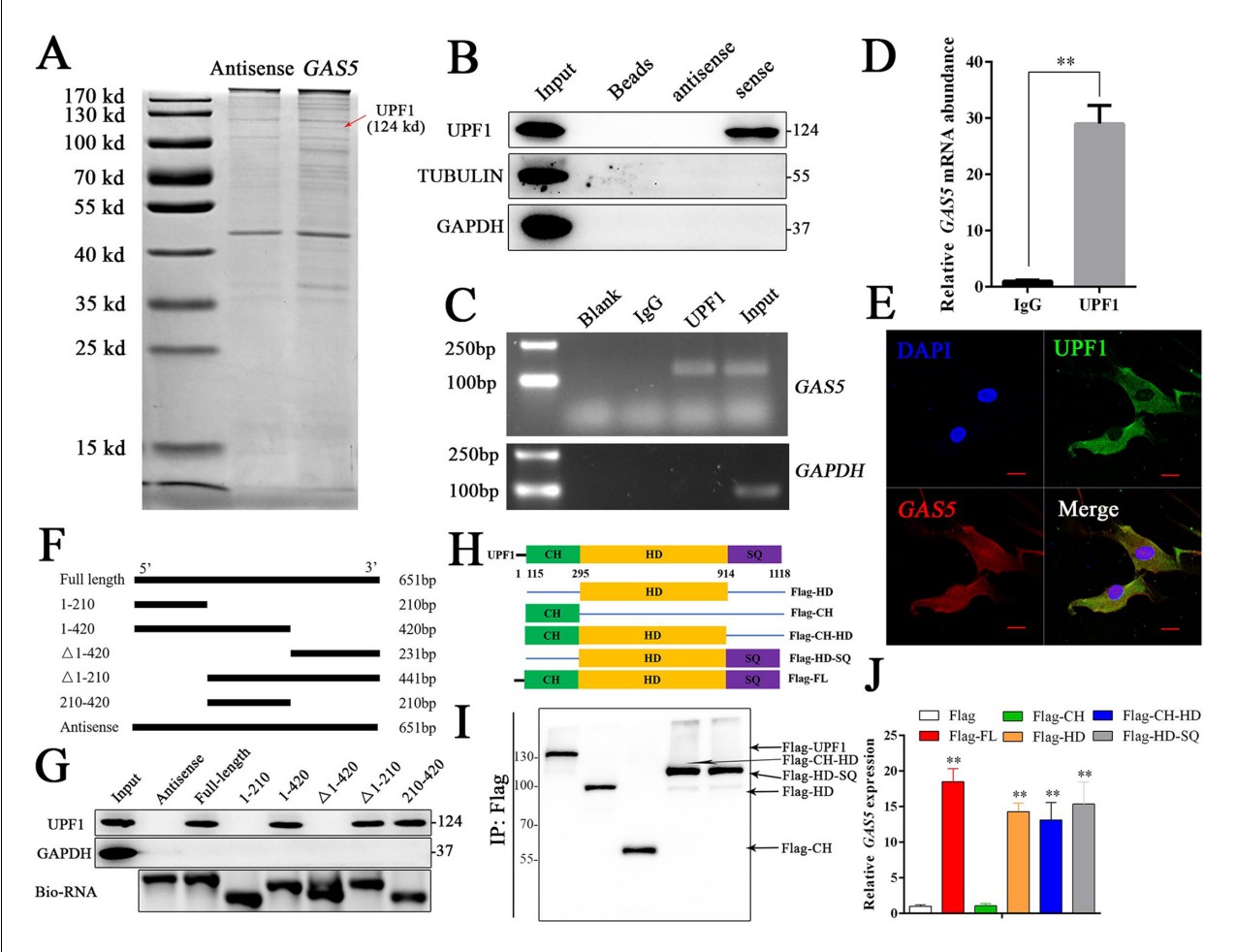

**Figure 4.** *GAS5* functions by interacting with the *UPF1* protein. (**A**) Coomassie brilliant blue staining of RNA pull-down proteins in the *GAS5* sense or antisense group. (**B**) The interaction between *GAS5* and UPF1 was confirmed by Western blotting. (**C, D**) RIP assay analysis using the anti-UPF1 antibody revealed that *GAS5* interacted with UPF1 in BMSCs, with GAPDH as a negative control. (**E**) *GAS5* RNA FISH was performed in BMSCs and showed that *GAS5* could partly colocalize with UPF1. Scale bar, 50 µm. (**F**) *GAS5* truncation sequence diagram. (**G**) RNA pull-down assay with *GAS5* truncation sequences and analyzed by Western blot. (**H**) UPF1 truncation diagram in a Flag tail. (**I**) Western blot analysis of the expression efficiency of UPF1 truncations. (**J**) RIP assay used UPF1 truncations and was analyzed by qRT-PCR. Each cellular experimental group was repeated at least three times.

The online version of this article includes the following figure supplement(s) for figure 4:

**Figure supplement 1.** QRT-PCR analysis of the *GAS5* adjacent genes in *GAS5* knockdown conditions.

**Figure supplement 2.** RNA fold model of *GAS5*.

GAS5-OE group, and the catenin and ERK pathways were impervious to *GAS5* expression. Further exploration of the TGF-β/SMAD1/5/8 pathway revealed that SMAD7, an inhibitor of SMAD1/5/8 (*Miyazawa and Miyazono, 2017*), was increased in the *GAS5* siRNA group and decreased in the *GAS5* overexpression group (*Figure 5B and C*). Other molecular factors in the SMAD1/5/8 pathways, such as BMP2, BMP4, BMP7 and SMAD6, were unchanged (*Figure 5—figure supplement 2A–F*). These results determined that SMAD7 in the SMAD1/5/8 signaling pathway was the target of the *GAS5*/UPF1 complex.

The RNA helicase UPF1 is one of the central molecules in nonsense-mediated mRNA decay (NMD) and Staufen-mediated mRNA decay (*Plank and Wilkinson, 2018*). UPF1 knockdown could increase the expression of *SMAD7* as well as inhibit the activation of the SMAD1/5/8 pathway (*Figure 5D–E*). Based on the function of UPF1, a key molecule in mRNA nonsense-mediated decay, we hypothesized that *GAS5* could affect *SMAD7* mRNA stability by binding to UPF1 in BMSCs. First,

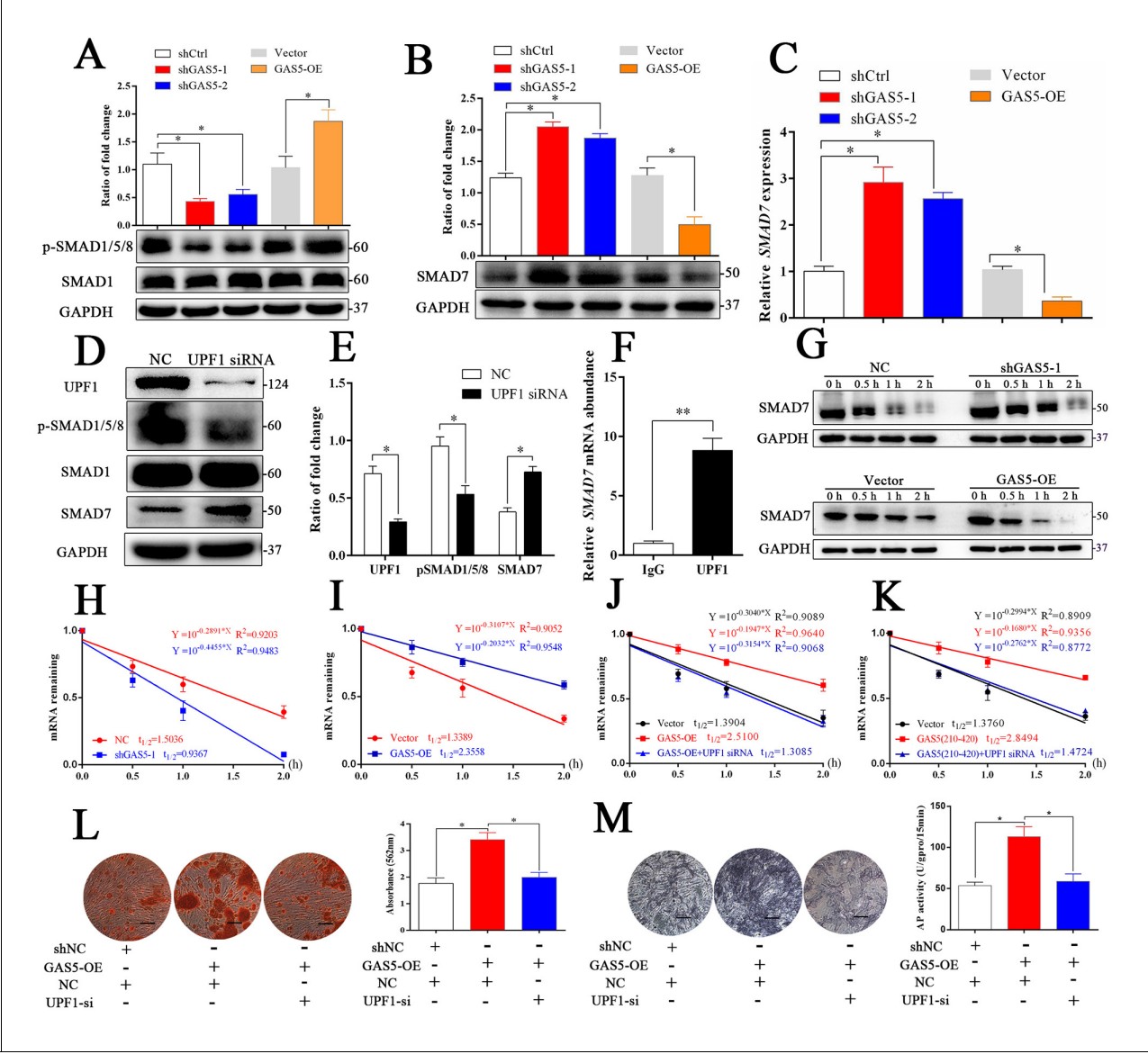

**Figure 5.** *GAS5* interacts with *UPF1* to accelerate *SMAD7* decay. (**A**) Western blot analysis of the activation pathway (pSMAD1/5/8, SMAD1) in *GAS5* knockdown or overexpression conditions. (**B**) SMAD7 was regulated by *GAS5* knockdown or overexpression by western blot. (**C**) Relative *SMAD7* mRNA levels in *GAS5* knockdown or overexpression cells tested by qRT-PCR. (**D, E**) Western blot analysis and quantification of pSMAD1/5/8/SMAD1 and SMAD7 in the UPF1 knockdown and control groups. (**F**) RIP assay with anti-UPF1 showed the binding of UPF1 and *SMAD7* mRNA. (**G**) Western blot analysis in the *GAS5* knockdown, overexpression and control groups after actinomycin D treatment. NC means the negative control siRNA transfected group. It was also be treated by Actinomycin D but as the control of *GAS5* shRNA-1. (**H**) Relative *SMAD7* mRNA expression in the *GAS5* knockdown and control groups by actinomycin D treatment tested by qRT-PCR. (**I**) Relative *SMAD7* mRNA expression in the *GAS5* overexpression and control groups by actinomycin D treatment tested by qRT-PCR. (**J, K**) Relative *SMAD7* mRNA expression by actinomycin D treatment under the condition of *UPF1* siRNA, overexpression of a truncated form of *GAS5* or *GAS5* full-length. (**L**) ARS staining and quantification in the *GAS5* overexpression and *UPF1* siRNA groups. Scale bar, 500 μm. (**M**) ALP staining and ALP tests in the *GAS5* overexpression and *UPF1* siRNA groups. Scale bar, 500 μm. Each cellular experimental group was repeated at least three times.

The online version of this article includes the following source data and figure supplement(s) for figure 5:

**Source data 1.** Original images of alizarin red staining and ALP staining in experiments of 'GAS5 interacts with UPF1 to accelerate SMAD7 decay'.

**Figure supplement 1.** Western blot analysis of the pathway.

**Figure supplement 2.** Western blot analysis of the molecular in BMP pathway.

we determined that UPF1 could bind to *SMAD7* mRNA by a RIP assay (*Figure 5F*). Next, we tested the mRNA abundance after the addition of actinomycin D, which is an inhibitor of DNA transcription, and found that the *SMAD7* mRNA decay speed was slower in the shGAS5 group but faster in the OE-GAS5 group at both the protein and RNA levels (*Figure 5G–I*). *UPF1* knockdown completely blocked the effect of *GAS5* on *SMAD7* mRNA decay speed (*Figure 5J*). Furthermore, the region of *GAS5 (210–420 bp)* could also enhance the speed of *SMAD7* mRNA decay and was blocked by UPF1 knockdown (*Figure 5K*). Knocking down UPF1 expression obviously blocked the function of *GAS5* in the regulation of osteoblast differentiation, as shown by ARS and ALP assays (*Figure 5L–M*). These results indicated that *GAS5* positively regulated the osteoblast differentiation by interacting with UPF1 to accelerate *SMAD7* mRNA decay.

## Gas5 heterozygous mice exhibit an osteoporotic phenotype and impaired bone repair capacity

To further explore the function of *GAS5* in bone metabolism, we constructed *Gas5* knockout mice. However, we used *Gas5* heterozygotes (*Gas5$^{+/-}$*) instead because all homozygous *Gas5$^{-/-}$* mice unexpectedly died during the embryonic period. QRT-PCR analysis and DNA agarose gel electrophoresis proved that *Gas5* partly deletion was successful (*Figure 6—figure supplement 1A–B*). And *Gas5* in cancellous bone of *Gas5$^{+/-}$*mice decreased compared with the WT mice (*Figure 6—figure supplement 1D*). The heterozygous *Gas5$^{+/-}$* mice survived normally and showed no changes in size, weight or appearance compared with their WT littermates (*Gas5$^{+/+}$*). However, the *Gas5$^{+/-}$* mice had less femoral cancellous bone than the WT mice, as shown by micro-CT and its three-dimensional reconstruction results (*Figure 6A*). In addition, H and E staining and *Bglap* immunohistochemical staining showed that bone trabeculae were decreased in *Gas5$^{+/-}$* mice (*Figure 6B*, *Figure 6—figure supplement 2*). In addition, bone analysis from micro-CT showed that BV/TV, trabecular thickness, and trabecular number were decreased, and BSA/BV and trabecular spacing were increased in *Gas5$^{+/-}$* mice compared with in WT mice (*Figure 6C–G*). As shown in *Figure 6H*, decreased *Bglap* levels in the sera of the *Gas5$^{+/-}$* group were observed compared with those in the WT group. These results demonstrated that *Gas5$^{+/-}$* mice showed osteoporosis-like manifestations. The BMSC of *Gas5$^{+/-}$* and WT group were isolated and induced into osteoblast differentiation. The mRNA expression of *Gas5*, *Upf1* and *Smad7* in BMSC of *Gas5$^{+/-}$* and WT mice were detected. *Gas5* was decreased in *Gas5$^{+/-}$* mice while *Smad7* was increase compared with WT mice. However, the level of *Upf1* mRNA has no significant change in two groups (*Figure 6—figure supplement 1C*). ARS and ALP staining showed that *Gas5$^{+/-}$* mouse BMSCs manifested a decreased osteoblast differentiation ability (*Figure 6I–L*). Furthermore, the detection of proteins isolated from the tibia indicated that the Smad1/5/8 pathway, rather than the catenin and ERK signaling pathways, was significantly lower in the *Gas5$^{+/-}$* group (*Figure 6M–N*, *Figure 6—figure supplement 3*). To determine the bone repair ability, a skull defect model was constructed using *Gas5$^{+/-}$* mice. Using micro-CT, we found that the bone repair area on cranial bone was lower in *Gas5$^{+/-}$* mice than in WT mice (*Figure 6O–P*). Collectively, these results suggest that *Gas5$^{+/-}$* mice exhibit an osteoporotic phenotype and impaired bone repair capacity.

## Systemic transfection (tail vein injection) of Gas5-overexpressing adenoviruses alleviated bone loss in osteoporosis

To investigate the therapeutic effects of *Gas5* on osteoporosis, we induced osteoporosis in mice with dexamethasone (DXMS) and injected *Gas5*-overexpressing adenoviruses or negative control. After *Gas5*-overexpressing adenovirus injection, the bone mass was alleviated as determined by micro-CT (*Figure 7A*) as well as by H and E staining and *Bglap* immunohistochemical staining (*Figure 7B*, *Figure 7—figure supplement 1*). Bone analysis from micro-CT showed that the BV/TV, trabecular thickness and trabecular number were increasing in the *Gas5* adenovirus group supplemented with drugs and had decreased BSA/BV and trabecular spacing compared with the negative control (*Figure 7C–G*). Taken together, these results suggest that *Gas5* may be a promising therapeutic target in osteoporosis.

## Discussion

Osteoporosis now needs more effective osteogenesis-promoting targets, as it currently has a limited therapeutic options (*Berry et al., 2019*; *Eastell et al., 2016*). Therefore, exploration of the specific

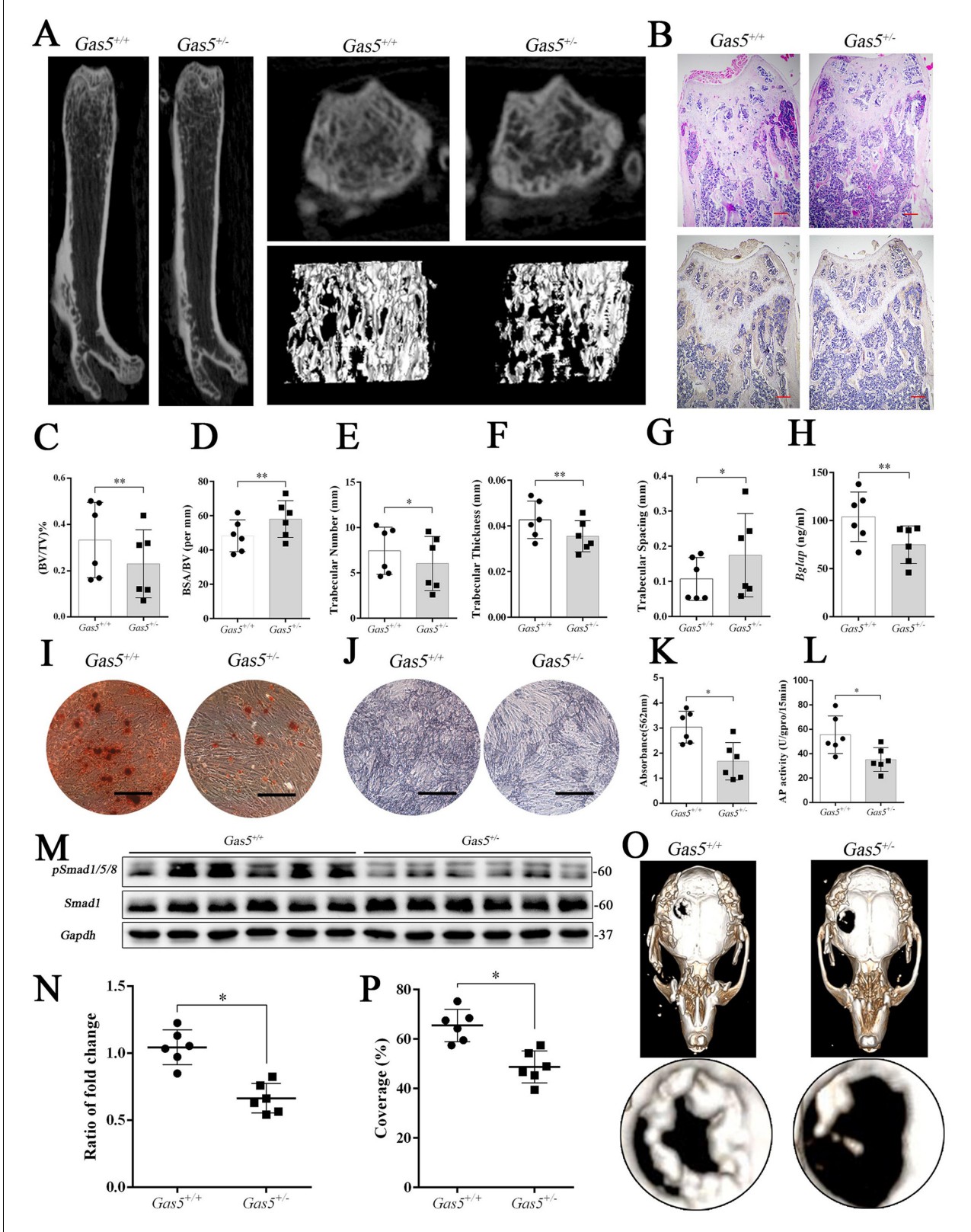

**Figure 6.** *Gas5* heterozygous mice exhibit an osteoporotic phenotype and impaired bone repair capacity. (**A**) Representative micro-CT analysis of the three-dimensional reconstruction graph for *Gas5*⁺/⁻ (n = 6) and WT mice (n = 6). (**B**) Representative H and E staining (upper) and *Bglap* immunohistochemical staining (down) of the terminal femur of *Gas5*⁺/⁻ (n = 6) and WT mice (n = 6). Scale bar, 500 μm. (**C, D, E, F, G**) BV/TV, BA/BV,

*Figure 6 continued on next page*

*Figure 6 continued*

trabecular thickness, trabecular number and trabecular spacing analysis of *Gas5*$^{+/-}$ (n = 6) and WT mice (n = 6). (H) Serum *Bglap* detection by ELISA in *Gas5*$^{+/-}$ (n = 6) and WT mice (n = 6). (I, K) ARS staining ARS and quantification of MSCs from *Gas5*$^{+/-}$ (n = 6) and WT mice (n = 6) Scale bar, 500 μm. (J) ALP staining of MSCs from *Gas5*$^{+/-}$ (n = 6) and WT mice (n = 6). (L) ALP tests of MSCs from *Gas5*$^{+/-}$ (n = 6) and WT mice (n = 6). Scale bar, 500 μm. (M, N) Western blot analysis for pSmad1/5/8/Smad1 in protein isolated from tibia of *Gas5*$^{+/-}$ (n = 6) and WT mice (n = 6). (O) Micro-CT three-dimensional images of cranial defects in *Gas5*$^{+/-}$ (n = 6) and WT mice (n = 6). (P) Bone recovery area analysis in *Gas5*$^{+/-}$ (n = 6) and WT mice (n = 6). Each cellular experimental group was repeated at least three times.

The online version of this article includes the following figure supplement(s) for figure 6:

**Figure supplement 1.** Relative gene expression in mouse.

**Figure supplement 2.** The quantification of *Bglap* immunohistochemical staining.

**Figure supplement 3.** Western blot analysis of the pathway.

gene responsible for osteoporosis and identification of new therapeutic targets are especially important. Previously, we found that *GAS5* could negatively regulate the adipocyte differentiation (*Li et al., 2018*). Herein, in our research, we found that as a protective target, *GAS5* decreases in both bone tissue and BMSCs, a major origin of osteoblast, of patients with osteoporosis. The osteoblast differentiation could be positively regulated by *GAS5* both in vitro and in vivo. Mechanistically, *GAS5* could specifically interact with UPF1 to accelerate the decay of *SMAD7*, which in turn activates the SMAD1/5/8 signaling pathway. Furthermore, *Gas5* heterozygous mice showed decreased bone mass and an impaired bone repair capacity. Bone loss in the osteoporosis mouse model was improved by systemic transfection of *Gas5*-overexpressing adenoviruses. Considering all of these findings together, we concluded that *GAS5* is a critical molecule in the osteoblast and

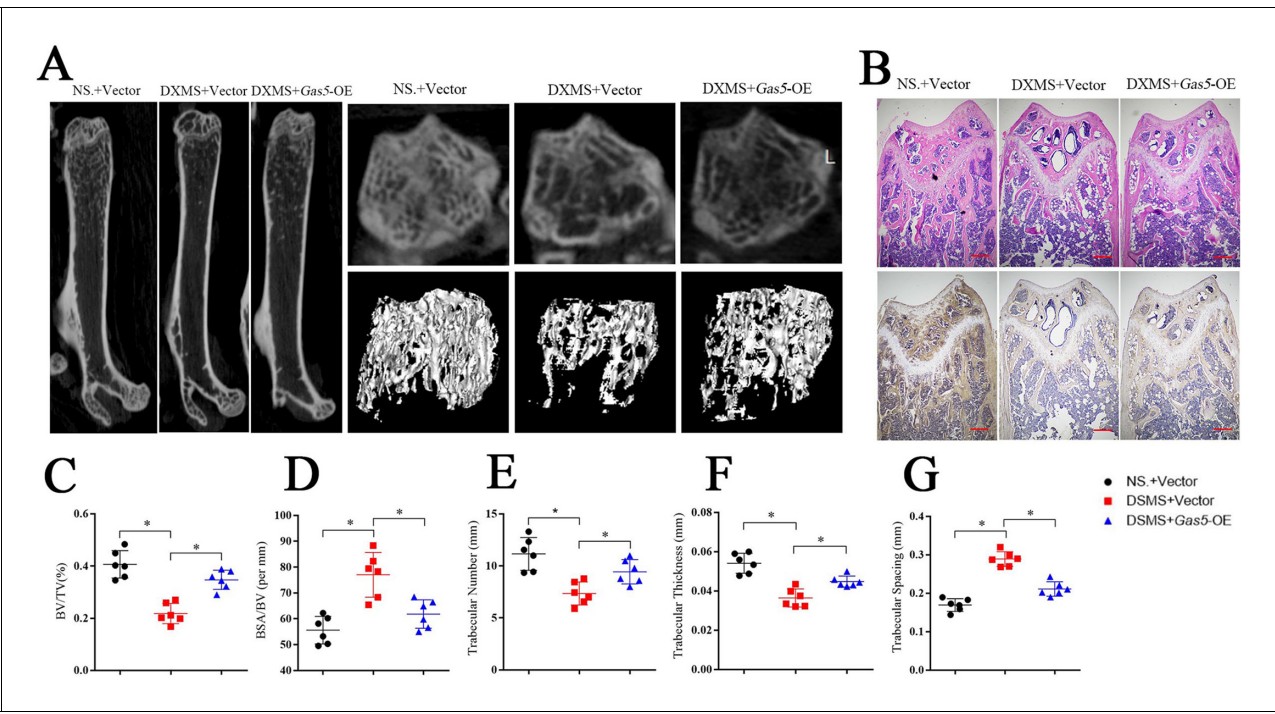

**Figure 7.** Systemic transfection (tail vein injection) of *Gas5*-overexpressing adenoviruses alleviated bone loss in osteoporosis. (A) Micro-CT analysis of the control mice (n = 6), osteoporosis mouse model (n = 6) or *Gas5*-overexpressing adenovirus-treated mice (n = 6). (B) H and E staining and *Bglap* immunohistochemical staining of the terminal femur of the control mice (n = 6), osteoporosis mouse model (n = 6) or *Gas5*-overexpressing adenovirus-treated mice (n = 6). Scale bar, 500 μm. (C, D, E, F, G) BV/TV, BA/BV, trabecular thickness, trabecular number, and trabecular spacing analysis for the control mice (n = 6), osteoporosis mouse model (n = 6) or *Gas5*-overexpressing adenovirus-treated group (n = 6). DXMS: dexamethasone.

The online version of this article includes the following figure supplement(s) for figure 7:

**Figure supplement 1.** The quantification of *Bglap* immunohistochemical staining.

adipocyte differentiation and contributes to the pathogenesis of osteoporosis, emphasizing its novel role in the diagnosis and treatment of this disease.

As the major originating cells of osteoblasts and adipocytes, BMSCs antagonistically perform osteoblast or lipoblast differentiation both in vivo and in vitro (*Li et al., 2018*). Physiologically, these differentiation abilities are balanced status and regulated by a variety of checkpoint molecules, including lncRNAs (*Grayson et al., 2015*; *Xie et al., 2019*). However, the abnormal expression of these checkpoint molecules could disrupt the differentiation balance, leading to abnormal bone metabolism as well as disease development (*Ma et al., 2018*; *Wu et al., 2016*; *Compston et al., 2019*). For example, the *lncRNA H19* could regulate osteogenesis by sponging *miR-675* and associating with cartilage regeneration, and abnormal *H19* expression was found in patients with osteoarthritis (*Huang et al., 2015*; *Liu et al., 2017*). Previously, we found that *GAS5* inhibited the adipocyte differentiation through the *miR-18a*/CTGF axis (*Li et al., 2018*). In our current research, we demonstrated that *GAS5* expression was positively correlated with the osteoblast differentiation, as shown by ARS and ALP assays, as well as critical osteogenesis marker expression. Inhibiting or overexpressing *GAS5* accordingly decreased or increased the osteoblast differentiation both in vitro and in vivo. Collectively, these in vitro and in vivo results demonstrated that *GAS5* was another critical checkpoint molecule for the osteoblasts and adipocytes differentiation by promoting osteogenesis and inhibiting adipogenesis.

Recent research found that *GAS5* could regulate osteogenic differentiation in mouse mesenchymal stem cells (*Wang et al., 2019*). While this result confirmed the role of *GAS5* in osteoblast differentiation, the physiological and pathological conditions as well as the molecular mechanisms are different between humans and mice. Therefore, it is important to further explore the detailed mechanisms of *GAS5* in the osteoblast differentiation of human. Critical signaling pathways for osteoblast differentiation, including SMAD1/5/8 (*Grafe et al., 2018*), catenin (*Kim et al., 2015*), and ERK (*Ye et al., 2018*) were detected, but only SMAD1/5/8 was regulated by *GAS5*. Furthermore, SMAD7, an inhibitory protein for the SMAD1/5/8 signaling pathway rather than for other molecules, was proven to be the target of *GAS5* in our study. Studies have found that SMAD7 inhibits the SMAD1/5/8 pathway and negatively regulates the osteoblast differentiation (*Li et al., 2017*; *Miyazawa and Miyazono, 2017*). Surprisingly, we found that *GAS5* could not directly interact with SMAD7. Instead, *GAS5* showed a strong affinity for and interactions with UPF1, as shown by a series of results in our research.

UPF1 is a key molecule in mRNA nonsense-mediated decay (NMD) (*Dehecq et al., 2018*). Recent studies indicated that UPF1 knockdown could inhibit the proliferation of colon cancer cells through the NMD mechanism (*Bokhari et al., 2018*). Studies have confirmed that lncRNAs can combine with different functional proteins to exhibit various effects. For instance, lncRNAs can enhance protein recruitment toward the associated target (*Chen et al., 2019*; *Yari et al., 2019*) or can strengthen/weaken the binding force of proteins and their targets (*Liu et al., 2018*; *Wang et al., 2020*). In this research, we demonstrated for the first time the role of *GAS5* in RNA NMD of BMSCs via UPF1. By combining and collecting UPF1, *GAS5* accelerated the degradation of SMAD7 and therefore promoted osteoblast differentiation.

Further exploration in our studies indicated that *GAS5* binds to the HD domain of UPF1. The HD domain was the key functional area in UPF1-associated NMD. Mutations of the HD domain disrupt the RNA helicase activity of UPF1 and perturb the function of UPF1 in NMD (*Dehecq et al., 2018*). Our results suggested that the binding of *GAS5* and UPF1 may strengthen the binding force of UPF1 and its target mRNA *SMAD7*. We speculated that in osteoblast differentiation, the increased expression of *GAS5* combined with UPF1 strengthened the binding of UPF1 and *SMAD7*, leading to the accelerated degradation of *SMAD7* and activation of the *SMAD1/5/8* pathway, ultimately promoting osteoblast differentiation.

Decreased bone mass and increased fat formation in bone are the main histopathological manifestations of osteoporosis (*Compston et al., 2019*; *Eastell et al., 2016*; *Infante and Rodríguez, 2018*), which indicates an imbalance between the osteoblast and adipocyte differentiation in osteoporosis patients. Many studies have confirmed that BMSCs in osteoporosis patients possess a decreased osteogenic ability but an enhanced adipogenic ability (*de Paula and Rosen, 2020*; *Pagnotti et al., 2019*), which is an important reason for the abnormal bone mass decrease and fat accumulation in bone marrow of osteoporosis patients. However, whether *GAS5* participated in these processes was not clearly clarified. In our research, we found that the level of *GAS5* expression

was decreased in the bone tissue and BMSCs of osteoporosis patients, indicating that *GAS5* may be related to the progression of osteoporosis. To confirm this hypothesis, we constructed a global *Gas5* knockout mouse model to explore the role of *GAS5* in osteoporosis. However, global *Gas5* knockout caused unexpected lethality in the embryonic period, which indicated that *Gas5* was indispensable in the development of the embryo. Interestingly, the osteoporotic phenotype, including lower bone mass, was macroscopically and microscopically observed in surviving *Gas5* heterozygous mice, as shown by both micro-CT and histological staining, compared with in WT mice. Combining the above results, we speculated that decreased *Gas5* levels may change heterozygote bone mass by regulating the osteoblast differentiation. To further confirm the abnormal osteogenic ability of *Gas5* heterozygous mice, we constructed a cranial defect model and found that its bone repair ability was significantly weakened, which further confirmed the dysfunction of osteoblast differentiation in *Gas5* heterozygous mice. However, the mechanism of cranial bone formation may not be analogous to what is observed for bone formation and or bone healing in the long bone (*Katsimbri, 2017*; *Catala et al., 2019*). We do not know exactly how *Gas5* would impact long bone healing and that this is a needed future direction.

Currently, the treatment of osteoporosis is currently mainly aimed at osteoclasts (*Ensrud and Crandall, 2017*; *Roux and Briot, 2018*), and the new therapy with *teriparatide* has shown a more curative effect in osteoporosis (*Minisola et al., 2019*), which suggests that treatment options directed at osteoblasts are of great significance. Adenoviruses containing lncRNAs were used to treat mice and showed a significant therapeutic effect in many diseases, including ischemic myocardial injury (*Cai et al., 2019*) and cardiac regeneration (*Ponnusamy et al., 2019*). To clarify the possible role of *Gas5* in osteoporosis treatment, we supplemented *Gas5*-overexpressing adenoviruses in an osteoporosis mouse model based on the preceding results. Micro-CT, H and E staining, and *Bglap* immunohistochemical staining showed that the bone loss in the *Gas5* overexpression group was more alleviated than the bone loss in the control group. These results indicate that *Gas5* not only participates in the pathogenesis of osteoporosis but also may be a promising therapeutic target.

In conclusion, we found that *GAS5* acts as a protective target in osteoporosis. The protective function of *GAS5* was exhibited by promoting the osteoblast differentiation by the UPF1/SMAD7 axis. Targeting *GAS5* may greatly contribute to both the diagnosis and treatment of osteoporosis. However, several limitations still exist in our study. For example, more convincing evidence should be obtained by conducting experiments with an osteoblast-specific *Gas5* conditional knockout mouse. Further studies should be performed to address these issues in the future.

## Materials and methods

### Animals

The use plan of all involving animals (including the type and number of mice, the specific operation plan, and the principle of optimal substitution, etc.) was proposed, and the final plan was determined after internal discussion in the research group. The animal plan finally approved by the Animal Use and Care Committee of the Eighth Affiliated Hospital of Sun Yat-sen University. $Gas5^{+/+}$ and $Gas5^{+/-}$ mice (B6/JNju-Gas5em1Cd1152/Nju, pure C57BL/6J background) were purchased from GemPharmatech Co., Ltd. (Nanjing, China). Male and female $Gas5^{+/+}$ and $Gas5^{+/-}$ mice were euthanized at 3 and 6 months of age.

### Cell isolation and culture

Human BMSCs from 15 volunteers were isolated from bone marrow after informed consent was obtained in Center for Biotherapy, Sun Yat-sen Memorial Hospital. Briefly, bone marrow was extracted from the posterior superior iliac spine under sterile conditions. BMSCs were purified, isolated, and cultured using our previously reported methods. BMSCs at passage two were used in the experiments. BMSC identity was confirmed by the immunophenotype profile. The cells used in experiments were positive for the BMSC surface antigens CD29-phycoerythrin (PE), CD34-allophycocyanin (APC), and CD44-fluorescein isothiocyanate (FITC) and negative for CD45-FITC, CD105-FITC, and HLA-DR-PE. The source of BMSCs for *GAS5* location, function and mechanism exploration in BMSCs differentiation were from these 15 volunteers. Human tissue was obtained from healthy

donors after informed consent was obtained. The MSCs of normal control, hip dysplasia, and osteo-porosis patients was isolated in the above method. The study was approved by the ethics committee of the Eighth Affiliated Hospital of Sun Yat-sen University (Shenzhen, People's Republic of China). After explaining in detail for the possible risks and importance of the research, as well as informing methods of privacy protection, we obtained the informed consent and consent publish signatures of all patients or normal donors.

## Osteoblast differentiation and identification

For osteoblast differentiation, BMSCs were cultured in osteoblast differentiation medium (DMEM containing 10% FBS, 0.1 μM dexamethasone, 10 mM β-glycerol phosphate, 50 μM ascorbic acid, 100 IU/ml penicillin, and 100 IU/ml streptomycin) for 0–14 days. The medium was replaced every 3 days. The osteoblast differentiation was identified by alizarin red S (ARS) and alkaline phosphatase (ALP) assays as follows.

ALP staining was performed according to the manufacturer's instructions on day 10. Briefly, the cells were washed with PBS three times for 5 min, and an appropriate amount of BCIP/NBT staining liquid was added. The samples were incubated at room temperature in the dark for 20 min until the color developed. The working fluid was removed, and the cells were washed with distilled water twice to stop the reaction, and pictures were taken under the microscope.

ARS staining was performed as described in our previous study. The cells were fixed with paraformaldehyde for 30 min and washed with PBS three times. Alizarin red staining solution was added, and the cells were stained for 15 min at room temperature. The cells were washed again with PBS three times, and photos were taken under the microscope. The cells were destained for 1 hr at room temperature using 10% cetylpyridinium chloride monohydrate for ARS quantification. Afterwards, 200 μL of the liquid was transferred to a 96-well plate, and the spectrophotometric absorbance was measured at 562 nm.

## RNA isolation, reverse transcription, and qRT-PCR analyses

Total RNA was isolated from BMSCs with an RNA-Quick Purification Kit (ESscience, Guangzhou) and transcribed into complementary DNAs using a PrimeScript RT Reagent Kit (TaKaRa, Dalian) according to the manufacturer's instructions. Real-time PCR was performed using SYBR Green Premix Ex Taq (TaKaRa, Dalian). The data were standardized based on *GAPDH* expression, and the $2^{-\Delta\Delta Ct}$ method was used to analyze the data and determine the relative expression of each gene. The forward and reverse primers for each gene are listed in *Supplementary file 1*.

## Western blot and antibodies

Cells and tissues were lysed in RIPA lysis buffer containing a protease inhibitor cocktail. Protein lysates were separated by SDS polyacrylamide gel electrophoresis (SDS-PAGE) and transferred onto PVDF membranes. Then, the membranes were blocked with BSA and incubated with primary antibodies against UPF1 (1:5000, ab109363, Abcam), SMAD7 (1:2000, MAB2029, RD system), SMAD1/5/9 (1:1000, 13820S, Cell Signaling Technology), SMAD1 (1:1000, 6944S, CST), beta-Catenin (1:1000, 8480S, CST), Nonphospho (Active) β-Catenin (1:1000, 19807S, CST), Phospho-p44/42 MAPK (Erk1/2) (1:1000, 4370, CST), p44/42 MAPK (Erk1/2) (1:1000, 4695, CST), BMP2 (1:1000, ab14933, Abcam), BMP4 (1:1000, ab124715-40 μl, Abcam), BMP6 (1:1000, ab155963-40 μl, Abcam), BMP7 (1:1000, ab129156-40 μl, Abcam), SMAD6 (1:1000, ab80049, Abcam), GAPDH (1:5000, AF0006, Beyotime), overnight at 4°C. The protein signals were detected using chemiluminescent reagents (Millipore) according to the manufacturer's instructions.

## Nuclear and cytoplasmic fractionation and agarose gel electrophoresis

Cytoplasmic and nuclear RNA was extracted as described previously according to the manufacturer's instructions (PARIS, Part Number AM1921). Cells were lysed in lysis solution and isolated in cell disruption and cell fractionation buffer. qRT-PCR was performed to analyze the RNA extracted from each of the fractions. The data were analyzed to determine the nuclear and cytoplasmic levels of each RNA.

PCR products were collected with DNA loading buffer and separated on 1.2% agarose containing 0.01% Gel Red (Biotium). Thereafter, DNA products were detected by UV light and photographed for observation.

## RNA pull-down assay and mass spectrometry

*GAS5* and its mutants were cloned into the pcDNA3.1(+) vector. Linearized pcDNA3.1(+) expressing *GAS5*- or *GAS5*-deletion sequences was used as a template to synthesize biotinylated *GAS5*, *GAS5* antisense or mutant sequences by the TranscriptAid T7 High-Yield Transcription Kit (Thermo Fisher Scientific). A Pierce RNA 3′ End Desthiobiotinylation Kit was used to attach a single desthiobiotinylated cytidine bisphosphate to the 3′ end of the RNA strand. After biotin labeling, the Pierce Magnetic RNA-Protein Pull-Down Kit (Thermo Fisher Scientific) was used for the RNA pull-down assay following the manufacturer's suggestions. Labeled RNA was captured using 50 μL of streptavidin magnetic beads in RNA Capture Buffer for 30 min at room temperature. RNA pull-down specificity was assessed by western blotting. The RNA-binding proteins were sequenced and identified by LC-MS as described by BBI Life Sciences Corporation.

## RNA immunoprecipitation (RIP)

RIP was performed using a previously described method (*Li et al., 2018*). In brief, $5 \times 10^6$ cells were harvested by RIP lysis buffer and then incubated with RIP buffer containing magnetic beads conjugated to anti-UPF1 and anti-Flag antibodies. Purified rabbit IgG was used as a negative control. A positive control anti-SNRNP70 antibody was used for the RIP procedure. Then, immunoprecipitated RNAs were isolated and purified for quantitative PCR analysis to detect the presence of the target lncRNA.

## Plasmid, lentivirus construction, and infection

Three *GAS5* and *UPF1* siRNAs were designed and synthesized by GenePharma (Shanghai, China). Details of the sequences are shown in *Supplementary file 2*. The best knockdown efficiency siRNA was selected to build the short hairpin RNA (shRNA) and construct the lentivirus by OBiO Technology (Shanghai) Corp., Ltd. *GAS5* lentiviruses, *Gas5*-overexpressing adenoviruses and the plasmids UPF1-Flag, CH-Flag, HD-Flag, CH-HD-Flag, and HD-SQ-Flag were constructed and purchased from OBiO Technology (Shanghai). Completed lentiviruses and adenoviruses were used to transfect BMSCs and animals, respectively, for functional testing. BMSCs were incubated with the lentiviruses for 24 hr at a multiplicity of infection of 50.

## Bone formation of BMSCs in vivo

BMSCs were transfected with lentiviruses (shGAS5-1, shGAS5-2, negative control, GAS5-OE or the vector) were cultured in osteoblast medium for 7 days prior to the in vivo study. The cells ($5 \times 10^5$) were resuspended and planted onto 40 mg hydroxyapatite/tricalcium phosphate (HA/TCP) scaffold. Then, the scafflods were implanted subcutaneously into the backs of 8-week-old BALB/c homozygous nude (nu/nu) mice. The implants were removed after 8 weeks and fixed in 4% paraformaldehyde. H and E, Masson, and immunofluorescence staining were performed to analyze the new bone formation.

## Collection of human bone samples

The femoral head was acquired during the surgery after signing informed consent with all donors. Eight postmenopausal osteoporosis patients with femoral neck fractures and eight hip dysplasia patients requiring surgery and eight normal control patients with the accident required surgery were recruited in this research. The characteristics of the study subjects are presented in *Supplementary file 3*.

## Fluorescent in situ hybridization (FISH)

For RNA FISH, a RiboFluorescent In Situ Hybridization Kit (RiboBio company) was used to detect the RNA in BMSCs according to the manufacturer's instructions. In brief, cells in wells were fixed with 4% polyoxymethylene for 20 min, permeabilized and prehybridized in hybridization buffer and then hybridized at 55°C for 1 hr with a *GAS5* probe mix (RiboBio company). Stained samples were

mounted with Fluoromount-G for confocal imaging. Images were obtained using an LSM 5 Exciter confocal imaging system (Carl Zeiss).

## Osteoporosis mouse model construction and treatment

Glucocorticoid-induced C57BL/6J mice were used to establish an osteoporosis model (dexamethasone (DXMS) dosage: 2 mg/kg, hind leg muscle injection, three times a week for 8 weeks). A total of $5 \times 10^{11}$ *Gas5*-overexpressing adenoviruses or control adenoviruses was injected into the mice from the tail vein. Eight weeks later, the serum and femur of the mice were acquired for micro-CT and staining studies.

## Cranial defect experiment

Ten-week-old mice were anesthetized, and the skin was prepared, and the surgical area was strictly disinfected. The skin was incised with a scalpel, and the subcutaneous tissue was carefully peeled off. A 2.5 mm sterilized drill bit was used to drill the skull, and the position was controlled at the left posterior cranial fossa. When there was a sense of breakthrough, the skull piercing was stopped, and dura mater puncturing was avoided. The bone fragments were carefully removed, and the incision was sutured with a thin thread. The incision was disinfected, and the surgery mouse was placed in a new cage. Eight weeks later, the whole skull of the mice was acquired for micro-CT analysis.

## Enzyme-linked immunosorbent assay (ELISA)

Serum levels of osteocalcin (*Bglap*) were tested by ELISA according to the manufacturer's instructions (Mouse Osteocalcin ELISA Kit (Colorimetric)). First, 100 µl of Capture Antibody Solution was added into the wells. The plate was incubated overnight at 4°C, and the plates were blocked by dispensing 300 µl of reagent diluent into each well and incubating the plate for 60 min at room temperature. Then, 100 µl of diluted samples and standards was added to the appropriate wells, and the plate was covered. The plate was incubated for 2 hr at room temperature. Finally, 100 µl of Detection Antibody Solution was dispensed into each well. The plate was incubated for 2 hr at room temperature. Then, 100 µl of the Substrate Solution was added into each well. The plate was covered and incubated for 20 to 30 min at room temperature. After color development, the cover was removed, and 50 µl of Stop Solution was dispensed into each well to stop the enzymatic reaction. The microwell absorbances were read immediately at 450 nm. The mouse serum was collected and diluted 1:20 for measurement of total *Bglap*.

## Microcomputed tomography (micro-CT) and histomorphometric analyses

Microcomputed tomography (micro-CT) was used to analyze the bone structure of the femur and skull. The harvested bone tissues were fixed in 4% polyoxymethylene for 2 days and then stored in 70% ethanol at 4°C before being processed. To analyze trabecular bone, images were acquired at an effective pixel size of 8.82 lm, a voltage of 80 kV, a current of 500 lA and an exposure time of 1500 ms in each of the 360 rotational steps. The bone volume/total vol (BV/TV), bone surface area/bone vol (BSA/BV), trabecular thickness, trabecular number and trabecular spacing were analyzed according to the guidelines. Two-dimensional and three-dimensional bone structure image slices were reconstructed.

## H and E, Masson, and immunohistochemical staining

The dissected tissues were fixed in 4% polyoxymethylene for 2 days and decalcified in 20% EDTA for 1 week before sectioning. Five-µm-thick slices were prepared on charged glass slides. H and E staining and Masson staining were performed according to the manufacturer's instructions (Beijing Solarbio Science and Technology Co., Ltd.). The slides were immersed in 10 mM citrate buffer (pH 7.5) and microwaved at 750 W for 30 min for antigen retrieval for immunohistochemical staining. After that, the samples were treated with 3% $H_2O_2$ for 20 min and blocked with 5% normal goat serum for 1 hr at room temperature. Primary antibodies were applied in blocking buffer or hybridization buffer at 4°C overnight. Standard DAB staining was performed for chromogenic detection of immunohistochemistry.

## Statistical analysis

Each cellular experimental group was repeated at least three times. All data are expressed as the mean ± standard deviation (SD). Statistical analysis was performed using SPSS 18.0 software (SPSS, Chicago, IL, USA). Statistical significance between *Gas5* heterozygous and wild-type (WT) mice was calculated using a paired two-tailed t test. Statistical differences were analyzed by unpaired two-tailed Student's t test for other comparisons between two groups. Analysis of variance (ANOVA) and appropriate post hoc analyses were used for comparisons of more than two groups. Correlations between the quantification of ARS staining, ALP tests, *ALP*, *RUNX2*, and *BGLAP* expression and *GAS5* expression in MSCs were examined using Pearson's correlation. A probability value of 0.05 or less was considered statistically significant.

## Acknowledgements

The authors thank American Journal Experts for providing an English language editing service, which was used for this manuscript.

## Additional information

### Funding

| Funder | Grant reference number | Author |
|---|---|---|
| National Natural Science Foundation of China | 81971518 | Zhongyu Xie<br>Peng Wang<br>Yanfeng Wu<br>Huiyong Shen |
| Fundamental Research Funds for the Central Universities | 19ykpy01 | Zhongyu Xie |
| National Natural Science Foundation of China | 81702120 | Zhongyu Xie<br>Peng Wang<br>Yanfeng Wu<br>Huiyong Shen |
| National Natural Science Foundation of China | 81871750 | Zhongyu Xie<br>Peng Wang<br>Yanfeng Wu<br>Huiyong Shen |
| National Natural Science Foundation of China | 81672097 | Zhongyu Xie<br>Peng Wang<br>Yanfeng Wu<br>Huiyong Shen |
| Guangdong Provincial Key R&D Programme | 2019B020236001 | Zhongyu Xie<br>Peng Wang<br>Huiyong Shen |

The funders had no role in study design, data collection and interpretation, or the decision to submit the work for publication.

### Author contributions

Ming Li, Conceptualization, Data curation, Formal analysis, Investigation, Methodology, Writing - original draft; Zhongyu Xie, Conceptualization, Formal analysis, Methodology, Writing - original draft, Project administration; Jinteng Li, Data curation, Formal analysis, Writing - original draft; Jiajie Lin, Data curation, Investigation, Methodology; Guan Zheng, Data curation, Formal analysis; Wenjie Liu, Data curation, Validation, Methodology; Su'an Tang, Software, Investigation; Shuizhong Cen, Software, Formal analysis, Investigation; Guiwen Ye, Formal analysis, Investigation, Methodology; Zhaofeng Li, Data curation, Methodology; Wenhui Yu, Data curation, Software, Formal analysis; Peng Wang, Conceptualization, Project administration; Yanfeng Wu, Validation, Project administration; Huiyong Shen, Resources, Funding acquisition, Project administration, Writing - review and editing

## Author ORCIDs

Huiyong Shen https://orcid.org/0000-0001-7104-3049

## Ethics

Human subjects: The study was approved by the ethics committee of the Eighth Affiliated Hospital of Sun Yat-sen University (approval number: 2018r010) and it was performed in strict accordance with the recommendations of ethics committee. After explaining in detail the possible risks and importance of the research, as well as informing methods of privacy protection, we obtained the informed consent and consent publish signatures of all patients or normal donors.

Animal experimentation: This study was performed in strict accordance with the recommendations in the Guide for the Care and Use of Laboratory Animals of Eighth Affiliated Hospital of Sun Yat-sen University. All procedures involving animals were approved by the Animal Use and Care Committee of the Eighth Affiliated Hospital of Sun Yat-sen University (approval number: SYSU-IACUC-2018-B10325).

## Decision letter and Author response

Decision letter https://doi.org/10.7554/eLife.59079.sa1
Author response https://doi.org/10.7554/eLife.59079.sa2

# Additional files

## Supplementary files

- Supplementary file 1. Primers of the analyzed genes.

- Supplementary file 2. The siRNA sequences of the analyzed genes.

- Supplementary file 3. Characteristics of the study subjects.

- Supplementary file 4. Characteristics of the 15 healthy donors from Center for Biotherapy, Sun Yat-sen Memorial Hospital.

- Supplementary file 5. Data of mass spectrometry.

- Transparent reporting form

## Data availability

The relevant data are available from Dryad (DOI: https://doi.org/10.5061/dryad.9cnp5hqfj). Primers of the analyzed genes (Supplementary file 1), the siRNA sequences of the analyzed genes (Supplementary file 2), characteristics of the study subjects (Supplementary file 3) and Characteristics of the 15 healthy donors (Supplementary file 4) can be found in the supplementary documents.

The following dataset was generated:

| Author(s) | Year | Dataset title | Dataset URL | Database and Identifier |
|---|---|---|---|---|
| Li M, Xie Z, Li J, Lin J, Zheng G, Liu W, Tang S, Cen S, Ye G, Li Z, Yu W, Wang P, Wu Y, Shen H | 2024 | GAS5 protects against osteoporosis by targeting UPF1/Smad7 axis in osteoblast differentiation | https://doi.org/10.5061/dryad.9cnp5hqfj | Dryad Digital Repository, 10.5061/dryad.9cnp5hqfj |

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
