## [Decision Letter]

**Acceptance summary:**

Congratulations, we are pleased to inform you that your article, "*GAS5* protects against osteoporosis by targeting UPF1/*Smad7* axis in osteoblast differentiation", has been accepted for publication in *eLife*.

It is increasingly appreciated that long non-coding RNAs (lncRNAs) play a significant role in bone biology and pathology. In this study, Lui et al., show that one such lncRNA, *GAS5*, plays a significant role in vitro in marrow stromal cell differentiation into mature osteoblasts capable of mineralizing matrix and increases osteoblast number in vivo. They then went on to show that *GAS5* impacts osteoblastogenesis via modulating *SMAD7* mRNA stability by *GAS5's* interaction with UPF1, a regulator of non-sense mediated decay. This work thereby greatly increases our understanding of the nuances of SMAD regulation in osteoblastogenesis.

**Decision letter after peer review:**

Thank you for submitting your article "*GAS5* protects against osteoporosis by targeting UPF1/*Smad7* axis in osteoblast differentiation" for consideration by *eLife*. Your article has been reviewed by two peer reviewers, and the evaluation has been overseen by a Reviewing Editor and Clifford Rosen as the Senior Editor. The following individual involved in review of your submission has agreed to reveal their identity: Yuji Mishina (Reviewer #1).

The reviewers have discussed the reviews with one another and the Reviewing Editor has drafted this decision to help you prepare a revised submission.

*GAS5* is a long non-coding RNA that has been previously shown to have a role in tumorigenesis, but is less well characterized in other physiologic systems. This team has previously shown that *GAS5* regulates adipogenesis and in this paper they expand their focus to the osteoblast. While previous studies have suggested a role for *GAS5* in regulating osteoblastogenesis, this paper greatly expands on this. These authors first show that *GAS5* is reduced bone from osteoporotic patients and they use a combination of in vitro and in vivo models demonstrate that *GAS5* positively regulates osteoblast differentiation. Using a cutting edge proteomic approach, they identify UPF1 as interactor with *GAS5*, and that this interaction impacts BMP signaling. The authors show that the low bone mass phenotype of *Gas5* heterozygous mice can be partially rescued by systemic expression of *GAS5* and thereby demonstrating how non-coding genes potentially could contribute to human disease. There are 8 essential things that have to be addressed in the revision, although only 1, 2, and 4 will require additional experiments or data acquisition and presentation.Essential revisions:

1) It is well known that *SMAD7* negatively regulates both BMP and TGF-β signaling, which have distinct functions on bone homeostasis. The authors need to measure levels of TGF-β signaling, i.e. checking the levels of phospho-SMAD2/3 when *GAS5* function is altered.

2) As shown in Figure 5—figure supplement 2F, *SMAD6* levels are not changed by *GAS5*. Does UPF1 bind to *SMAD6* mRNA less efficiently than *SMAD7* mRNA? Providing data (or make a case based on published information) to explain distinctive outcomes between two inhibitory Smads will further signify this article.

3) The cranial defect model is an interesting model from which we have learned a great deal about bone healing, the concern was raised during review that the mechanism of cranial bone formation may not be analogous to what is observed for bone formation and or bone healing in the long bone. The limitations of the cranial defect model needs to be carefully and completely presented in the Discussion. It was requested that a long bone fracture healing experiment be conducted and this would certainly strengthen the paper, but it is also recognized that this might not be possible at this time. Therefore, the inclusion of data showing bone formation in a long bone fracture model is suggested but not required. If a fracture healing experiment is not done, it should be stated in the Discussion that we do not know exactly how *GAS5* would impact long bone healing and that this is a needed future direction.

4) The mRNA expression levels of *Gas5* in cancellous bone tissue in *Gas5*^+/-^ mice should be determined if possible.

5) In the Introduction, a more complete presentation of prior studies of *GAS5* in osteogenic differentiation should be included.

6) Quantification of the data presented in Figures 2E, 3E, 6B and 7B is required.

7) It is requested that for Figure 6B, the image of HE staining of *Gas5*^+/-^ be changed for a clearer image.

8) Figure 4A created a lot of confusion. The reviewers noted that there are several bands more intense than UPF1 in the *GAS5* group. Why do the authors focus on UPF1? Do they know identities of other bands?

---

## [Author Response]

Essential revisions:1) It is well known that SMAD7 negatively regulates both BMP and TGF-β signaling, which have distinct functions on bone homeostasis. The authors need to measure levels of TGF-β signaling, i.e. checking the levels of phospho-SMAD2/3 when GAS5 function is altered.

Thank you for your valuable comment. We detected other molecules in the TGF-β pathway, such as SMAD2/3, SMURF2, SMURF1. The results of Western blot showed that SMAD2/3, SMURF2, SMURF1 were not affected by *GAS5* knock down or overexpression. These results further testified that *GAS5* regulated osteoblast differentiation in SMAD7-SMAD1/5/8 pathway. The results were added in Figure 5—figure supplement 1A-C.

2) As shown in Figure 5—figure supplement 2F, SMAD6 levels are not changed by GAS5. Does UPF1 bind to SMAD6 mRNA less efficiently than SMAD7 mRNA? Providing data (or make a case based on published information) to explain distinctive outcomes between two inhibitory Smads will further signify this article.

Thank you for your valuable suggestions. For this question, the binding RNA of UPF1 was tested again with the RIP kit by applied the IP antibody of UPF1. The results of qPCR showed that compared with IgG control, *SMAD7* mRNA increased 8.83 times, while *SMAD6* mRNA increased only 1.47 times (Author response image 1). It is indicated that mRNA of *SMAD7* is significantly stronger than *SMAD6* in combination with UPF1. Therefore, *GAS5* mainly affects the mRNA degradation of *SMAD7* and further affects the osteogenic differentiation of MSCs by binding with UPF1.

**Author response image 1. sa2fig1:** (A)The *SMAD7* mRNA abundance by RIP assay with anti-UPF1 antibody; (B) the *SMAD6* mRNA abundance by RIP assay with anti-UPF1 antibody.

3) The cranial defect model is an interesting model from which we have learned a great deal about bone healing, the concern was raised during review that the mechanism of cranial bone formation may not be analogous to what is observed for bone formation and or bone healing in the long bone. The limitations of the cranial defect model needs to be carefully and completely presented in the Discussion. It was requested that a long bone fracture healing experiment be conducted and this would certainly strengthen the paper, but it is also recognized that this might not be possible at this time. Therefore, the inclusion of data showing bone formation in a long bone fracture model is suggested but not required. If a fracture healing experiment is not done, it should be stated in the Discussion that we do not know exactly how GAS5 would impact long bone healing and that this is a needed future direction.

Thank you for your valuable suggestions. After literature search and reading, we agree that the skull and long bone repair principle is different. The cranial bone formation is mainly achieved through intramembranous osteogenesis. The cranial bone is a fibrous connective tissue where cells are differentiated from stromal cells. The stromal cells proliferate and then differentiate into osteoblasts. After that osteoblasts deposit collagen fibers and minerals in the bone plate, which enlarges the bone plate and promotes fracture healing (Catala, et al., 2019; Opperman, 2000). Long bones formation mainly through endochondral ossification. After a series of continuous and coordinated differentiation steps, stromal cells form cartilage plate, secrete alkaline phosphatase, and promote calcification of cartilage matrix. Osteogenic precursor cells deposit osteoid matrix in primary ossification center and then mineralize bone matrix to repair long bones (Hinton, et al., 2017; Katsimbri, 2017). It is true that there are differences between the two repair methods, and it is inappropriate to simply use skull defect repair to represent the long bone healing experiment. Due to time limitation and effect of COVID-19, we were hard to conduct the long bone repair experiment, which needs further discussion in the future. We have added the repair difference of skull and long bone, and the limitation of our research in the sixth paragraph of the Discussion.

4) The mRNA expression levels of Gas5 in cancellous bone tissue in Gas5^+/-^ mice should be determined if possible.

Thank you for your valuable suggestions. We euthanized *Gas5* knockout heterozygous mice and wild type control mice at week 12. Their femurs were collected. After rinsing the bone marrow cavity with syringe, the cancellous part was scraped. Due to the little amount, we collected the cancellous bone material from both sides of the same mouse together and performed RNA extraction using the RNA extraction method mentioned in the manuscript. The level of *Gas5* was detected after reverse transcription of RNA into cDNA. The results indicated that the expression of *Gas5* in cancellous bone of *Gas5* knockout heterozygous mice decreased compared with the wild type mice. The results have been shown in Figure 6—figure supplement 1D.

5) In the Introduction, a more complete presentation of prior studies of GAS5 in osteogenic differentiation should be included.

Thank you for your valuable suggestions. We have given a more detailed description of previous research on osteogenesis function of *GAS5* in the third paragraph of the Introduction section.

6) Quantification of the data presented in Figures 2E, 3E, 6B and 7B is required.

Thank you for your valuable suggestions. The quantification of *Bglap* immunohistochemical staining were analyzed by the software Image J 1.53a followed the plug-in of IHC-Profiler. The staining quantification for Figure 2E, Figure 3E has been added in Figure 2—figure supplement 2. The staining quantification for Figures 6B and 7B has been added in Figure 3—figure supplement 2.

7) It is requested that for Figure 6B, the image of HE staining of Gas5^+/-^ be changed for a clearer image.

We are sorry for the unclear picture. We redid the femur HE staining of the wild type and *Gas5* heterozygous knockout mice, and the unclear pictures have been replaced in Figure 6B.

8) Figure 4A created a lot of confusion. The reviewers noted that there are several bands more intense than UPF1 in the GAS5 group. Why do the authors focus on UPF1? Do they know identities of other bands?

Thank you for your valuable comment. After Coomassie bright blue staining for the binding proteins of *GAS5*, mass spectrometry analysis was carried out for the 4 significant bands. The results were showed in Supplementary file 5. By combing the mass spectrum results, associated with the influence of *GAS5* on TGF-β/SMAD pathway suggested earlier, and combined the previous research that UPF1 could affect the protein level of *SMAD7*. We believe that UPF1 is the most likely protein to bind *GAS5* for downstream function regulation. For other proteins, we screened two of them for functional analysis. However, neither of them has a positive effect on the SMAD1/5/8 pathway when they were knock down by two selected siRNA. The results were shown in Author response image 2.

**Author response image 2. sa2fig2:** Functional verification of 2 genes and 4 bands in the MMS. A) SMAD1/5/8 pathway changes after siRNA knockdown of the two selected proteins. B) Quantification of western blot for two proteins. C) The bands which selected for the mass spectrum (red arrow). RPN1: ribophorin I. ECH1: enoyl-CoA hydratase 1.